# Genome taxonomy of the genus *Neptuniibacter* and proposal of *Neptuniibacter victor* sp. nov. isolated from sea cucumber larvae

Rika Kudo[1], Ryota Yamano[1], Juanwen Yu[1], Shotaro Koike[1], Alfabetian Harjuno Condro Haditomo[1,2], Mayanne A. M. de Freitas[3], Jiro Tsuchiya[1], Sayaka Mino[1], Fabiano Thompson[3], Jesús L. Romalde[4], Hisae Kasai[1], Yuichi Sakai[5], Tomoo Sawabe[1] *

**1** Laboratory of Microbiology, Faculty of Fisheries Sciences, Hokkaido University, Hakodate, Japan, **2** Aquaculture Department, Faculty of Fisheries and Marine Sciences, Universitas Diponegoro, Semarang, Indonesia, **3** Laboratory of Microbiology, Biology Institute, Federal University of Rio de Janeiro (UFRJ), Rio de Janeiro, Brazil, **4** Departamento de Microbiología y Parasitología, CRETUS & CIBUS-Facultad de Biología, Universidade de Santiago de Compostela, Santiago, Spain, **5** Hakodate Fisheries Research, Hokkaido Research Organization, Local Independent Administrative Agency, Hakodate, Japan

* sawabe@fish.hokudai.ac.jp

**Data Availability Statement:** The GenBank accession number for the 16S rRNA gene sequence of the type strain PT1 is LC716006. The

## Abstract

A Gram-staining-negative, oxidase-positive, strictly aerobic rod-shaped bacterium, designated strain PT1$^T$, was isolated from the laboratory-reared larvae of the sea cucumber *Apostichopus japonicus*. A phylogenetic analysis based on the 16S rRNA gene nucleotide sequences revealed that PT1$^T$ was closely related to *Neptuniibacter marinus* ATR 1.1$^T$ (= CECT 8938$^T$ = DSM 100783$^T$) and *Neptuniibacter caesariensis* MED92$^T$ (= CECT 7075$^T$ = CCUG 52065$^T$) showing 98.2% and 98.1% sequence similarity, respectively. However, the average nucleotide identity (ANI) and *in silico* DNA-DNA hybridization (DDH) values among these three strains were 72.0%-74.8% and 18.3%-19.5% among related *Neptuniibacter* species, which were below 95% and 70%, respectively, confirming the novel status of PT1$^T$. The average amino acid identity (AAI) values of PT1$^T$ showing 74–77% among those strains indicated PT1$^T$ is a new species in the genus *Neptuniibacter*. Based on the genome-based taxonomic approach, *Neptuniibacter victor* sp. nov. is proposed for PT1$^T$. The type strain is PT1$^T$ (JCM 35563$^T$ = LMG 32868$^T$).

## Introduction

The genus *Neptuniibacter*, belonging to the family *Oceanospirillaceae*, was first proposed by Arahal et al. (2007) with *Neptuniibacter caesariensis*, which was isolated from surface water from the Eastern Mediterranean [1]. Currently, four species including *N. caesariensis* have been described in this genus: *Neptuniibacter halophilus*, isolated from a salt field in southern Taiwan [2], *Neptuniibacter marinus* and *Neptuniibacter pectenicola*, both from a scallop

whole genome sequence of the PT1 and Neptuniibacter halophilus LMG 25378T have been deposited to DDBJ/ENA/GenBank under the accession numbers AP025763, and AP027292-AP027293. Raw reads used for those genome assembly have been also deposited at GenBank under accession number DRA015857.

**Funding:** This study was supported by Kaken 19K22262. The funders had no role in study design, data collection and analysis, decision to publish, or preparation of the manuscript.

**Competing interests:** The authors have declared that no competing interests exist.

hatchery in Norway [3]. Members of the genus *Neptuniibacter* have been isolated from seawater and/or salt water, and the genus is characterized as being Gram-staining negative, motile, oxidase-positive, strictly aerobic rods requiring NaCl and/or sea salts [1–3].

In the process of collecting reference genomes to understand the structure, function, and dynamics of the sea cucumber microbiota [4–6], the strain PT1[T] was isolated from the larvae of *Apostichopus japonicus* [4]. The strain PT1[T] possesses similar sequences to those of the key Amplicon Sequence Variant (ASV0001-0004), which significantly increased its abundance during the pentactula and juvenile stages found in meta16S analyses of the early developmental stages in sea cucumber [4]. The strain PT1[T] could have previously unknown interactions with the host sea cucumber. Moreover, PT1[T] is the only isolate identified to the genus *Neptuniibacter* among 237 isolates obtained from the early life stage of *A. japonicus* [4]. Due to the absence of comprehensive studies on the genomic characterization of the genus *Neptuniibacter* and fewer animal-associated *Neptuniibacter* isolates, genome fundamentals can contribute not only to host-microbes interaction studies but also to genome taxonomy. Here we report the genome characterization of the newly described *Neptuniibacter* sp. strain PT1[T], and propose it as *Neptuniibacter victor* sp. nov. based on the genome taxonomy.

## Materials and methods

### Bacterial strains and phenotyping

The strain PT1[T] was isolated from the pentactula larvae of *A. japonicus* reared in a laboratory aquarium in July 2019 [4]. In brief, larvae were collected using 40 μm nylon mesh (FALCON Cell Strainer, Durham, USA), washed once with sterilized seawater, and then manually homogenized in 1 mL filter-sterilized natural seawater for 60 seconds. Ten-fold serial dilutions of the homogenate were cultured on 1/5 strength ZoBell 2216E agar plates [4]. Bacterial colonies were purified using the same agar plates. After purification, the isolate was stored at -80˚C suspended in glycerol supplemented 1/5 strength ZoBell 2216E broth. *N. halophilus* LMG 25378[T], *N. caesariensis* CECT 7075[T], *N. marinus* CECT 8938[T], *N. pectenicola* CECT 8936[T] were used for genomic and phenotypic comparisons against the strain PT1[T]. All strains were cultured on Marine agar 2216 (BD, Franklin Lakes, New Jersey, USA). The phenotypic characteristics were determined according to previously described methods and Traitar approach [5–10]. Motility was observed under a microscope using cells suspended in droplets of sterilized 75% artificial seawater [5, 6]. API20 NE (bioMérieux, Craponne, France) was also used to examine phenotypes according to Chen et al. [2].

Due to difficulties in the growth of PT1[T] in manually prepared marine Oxidative/Fermentative (OF), a salt requirement test, and synthetic basal marine media for evaluating carbon assimilations, we also used the *in silico* phenotyping tool, Traitar ver. 1.1.2 (Microbial Trait Analyzer) [11], instead of the experimental approach to compare predicted phenotypes based on genome sequences among *Neptuniibacter* species. This software is capable of predicting 67 phenotypic traits using Prodigal for gene prediction and Pfam family for annotation [11]. The software uses two prediction models, the phypat model (which predicts the presence/absence of proteins found in the phenotype of 234 bacterial species) and a combination of phypat +PGL models (which uses the information of phypat combined with the information of the acquisition or loss of protein families and phenotypes through evolution), to determine the phenotypic characteristics [11].

### Whole genome sequencing

Genomic DNAs of PT1[T] and *N. halophilus* were extracted from cells grown in Marine Broth 2216 using the Wizard genomic DNA purification kit (Promega, Madison, WI, USA)

according to the manufacturer's protocol. Genome sequencing was performed using both Oxford Nanopore Technology (ONT) MinION and Illumina MiSeq platforms. For the ONT sequencing, the library was prepared using Rapid Barcoding Sequence kit SQK-RBK004 (Oxford Nanopore Technologies, Oxford, UK) according to the standard protocol provided by the manufacturer. The library was loaded into a flow cell (FLO-MIN 106), and a 48-hour sequencing run with MinKNOW ver. 3.6.0 software was performed. Basecalling was carried out using Guppy ver. 4.4.1 (Oxford Nanopore Technologies). For Illumina sequencing, a 300 bp paired-end library was prepared using the NEBNext Ultra II FS DNA Library Prep Kit. The ONT and Illumina reads were assembled using Unicycler ver. 0.4.8 [12]. After checking quality value (QV) of these Illumina reads over 27 with no adaptor and barcode sequences assessed using FastQC program ver. 0.11.9 (https://www.bioinformatics.babraham.ac.uk/projects/fastqc/), these reads were used for genome assembly without preprocessing. Genome sequences of *N. caesariensis*, *N. marinus*, and *N. pectenicola* were retrieved from the NCBI database; the accession numbers are GCF_000153345.1, GCF_001597725.1, and GCF_001597735.1 [1, 3]. The whole genome sequences were annotated with DDBJ Fast Annotation and Submission Tool (DFAST) [13]. The complete genome sequences of PT1[T] and *N. halophilus* LMG 25378[T] acquired in this study were deposited in GenBank/EMBL/DDBJ under accession number AP025763 and AP027292-AP027293, respectively.

## Molecular phylogenetic analysis based on 16S rRNA gene nucleotide sequences

The full-length 16S rRNA gene sequence of strain PT1[T] was obtained from the complete genome sequence. The 16S rRNA gene nucleotide sequences of the type strains of the genus *Neptuniibacter* and other *Oceanospirillaceae* species were retrieved from NCBI databases. Sequences were aligned using MEGA X ver. 11.0.11 [14]. A phylogenetic model test and reconstruction of maximum likelihood (ML) tree were performed using the MEGA X ver. 11.0.11 [14, 15]. ML tree was reconstructed with 1,000 bootstrap replications using Kimura 2-parameter (K2) with gamma distribution (+G) and invariant site (+I) model. In addition, nucleotide similarities among strains were also calculated using the MEGA X software [14].

## Overall genome relatedness indices (OGRIs)

Overall genome relatedness indices (OGRIs) were calculated to determine the novelty of PT1[T] using same approach by Yamano et al. [5, 6]. Average nucleotide identities (ANIs) were calculated using the Orthologous Average Nucleotide Identity Tool (OrthoANI) software ver. 0.93.1 [16] using genomes of the PT1[T] and previously described *Neptuniibacter* type strains. *In silico* DNA-DNA hybridization (DDH) values were calculated using Genome-to-Genome Distance Calculator (GGDC) ver. 2.1 with formula 2, being the most robust against incomplete genomes [17]. Average amino acid identities (AAIs) were calculated and compared between PT1[T] and *Neptuniibacter* species (Table 1) using an enveomics toolbox [18].

## Multilocus sequence analysis (MLSA)

MLSA including concatenation of sequences and phylogenetic network reconstruction were performed using SplitsTree ver. 4.16.2 with the almost same setting [5–10, 19]. The sequences of four protein-coding genes (*mreB*, *recA*, *rpoA*, and *topA*) were obtained from the genome sequences of PT1[T], *N. halophilus* LMG 25378[T], *N. caesariensis* CECT 7075[T], *N. marinus* CECT 8938[T], *N. pectenicola* CECT 8936[T] and other related *Oceanospirillaceae* species (Table 1). These sequences were aligned using Clustal X ver. 2.1 [20]. Regions used for the network reconstruction in Fig 2 were 1–995, 1–1,041, 1–981, and 1–2,648 for *mreB*,

**Table 1. List of genomes used in this study.**

| Species | Strain | RefSeq/GenBank accession | Size (bp) | No. Of contigs | G+C content | MLSA | AAI |
|---|---|---|---|---|---|---|---|
| *Neptuniibacter victor* | PT1[T] | AP025763 (in this study) | 3,952,146 | 1 | 44.2% | + | + |
| *Neptuniibacter halophilus* | LMG 25378[T] | AP027292-AP027293 (in this study) | 4,113,600 | 2 | 53.3% | + | + |
| *Neptuniibacter caesariensis* | MED92[T] | GCF_000153345.1 | 3,924,755 | 6 | 46.0% | + | + |
| *Neptuniibacter marinus* | ATR 1.1[T] | GCF_001597735.1 | 3,454,191 | 117 | 42.8% | + | + |
| *Neptuniibacter pectenicola* | LFT 1.8[T] | GCF_001597725.1 | 3,631,894 | 203 | 45.7% | + | + |
| *Aliamphritea ceti* | RA1[T] | AP025282 | 5,213,210 | 1 | 47.1% | + | - |
| *Aliamphritea hakodatensis* | PT3[T] | AP025281 | 5,208,344 | 1 | 52.2% | + | - |
| *Aliamphritea spongicola* | MEBiC05461[T] | AP025283 | 4,964,135 | 167 | 51.5% | + | - |
| *Amphritea balenae* | JAMM1525[T] | GCF_014646975.1 | 4,671,747 | 18 | 47.7% | + | - |
| *Amphritea pacifica* | ZJ14W[T] | GCF_016924145.1 | 4,702,884 | 282 | 51.2% | + | - |
| *Amphritea atlantica* | M41[T] | AP025284 | 4,804,482 | 1 | 51.1% | + | - |
| *Amphritea japonica* | JAMM1866[T] | AP025761-AP025762 | 3,880,036 | 2 | 47.6% | + | - |
| *Amphritea opalescens* | ANRC-JH13[T] | GCF_003957515.1 | 4,124,452 | 46 | 48.5% | + | - |
| *Neptunomonas antarctica* | S3-22[T] | GCF_900156635.1 | 4,569,005 | 25 | 45.7% | + | - |
| *Neptunomonas phycophila* | Scap09 | GCF_013394205.1 | 3,976,465 | 1 | 45.5% | + | - |
| *Marinomonas arctica* | BSI20414 | GCF_014623465.1 | 4,540,024 | 1 | 44.5% | + | - |
| *Marinomonas mediterranea* | MMB-1[T] | GCF_000192865.1 | 4,684,316 | 1 | 44.1% | + | - |
| *Marinomonas posidonica* | IVIA-Po-181[T] | GCF_000214215.1 | 3,899,940 | 1 | 44.3% | + | - |
| *Oceanospirillum beijerinckii* | NBRC 15445[T] | GCF_000422425.1 | 5,325,321 | 99 | 47.8% | + | - |
| *Oceanospirillum maris* | ATCC 27509[T] | GCF_000422865.1 | 3,709,807 | 44 | 45.9% | + | - |
| *Oceanospirillum multiglobuliferum* | NBRC 13614[T] | GCF_900167095.1 | 3,512,709 | 46 | 45.4% | + | - |
| *Marinobacterium aestuarii* | ST58-10[T] | GCF_001651805.1 | 5,191,608 | 1 | 58.8% | + | - |
| *Marinobacterium litorale* | DSM 23545[T] | GCF_000428985.1 | 4,380,400 | 64 | 56.4% | + | - |
| *Marinobacterium mangrovicola* | DSM 27697[T] | GCF_004339595.1 | 4,979,947 | 15 | 57.1% | + | - |
| *Nitrincola lacisaponensis* | 4CA[T] | GCF_000691225.1 | 3,412,133 | 43 | 52.1% | + | - |
| *Nitrincola tapanii* | MEB 193[T] | GCF_008368715.1 | 2,793,747 | 19 | 50.8% | + | - |
| *Nitrincola tibetensis* | xg18[T] | GCF_003284585.1 | 4,001,852 | 54 | 46.1% | + | - |

+: used, -: not used.

*recA*, *rpoA*, and *topA*, as PT1[T] nucleotide sequence positions (GenBank accession number AP025763), respectively.

## Pan and core genome analyses

A total of five genomes, including two newly obtained in this study (PT1[T] and *N. halophilus* LMG 25378[T]) and three retrieved from the NCBI database (*N. caesariensis*, *N. marinus*, *N. pectenicola*) were used for pan- and core-genome analyses using the program anvi'o ver. 7 [21] based on previous studies [5, 6, 10, 22], with minor modifications. Briefly, contig databases of each genome were constructed by fasta files (anvi-gen-contigs-database) and decorated with hits from HMM models (anvi-run-hmms). Subsequently, functions were annotated for genes in a contigs database (anvi-run-ncbi-cogs). KEGG annotation was also performed (anvi-run-kegg-kofams). The storage database was generated (anvi-gen-genomes-storage) using all contigs databases and pangenome analysis was performed (anvi-pan-genome). The results were displayed (anvi-display-pan) and adjusted manually.

### *In silico* chemical taxonomy

The genes encoding key enzymes and proteins for the synthesis of fatty acids (Fas), polar lipids, and isoprenoid quinones were retrieved from the genome sequences of PT1[T] and four previously described *Neptuniibacter* species using *in silico* MolecularCloning ver. 7 (In Silico Biology, Yokohama, Japan) based on the same approach of Yamano et al. [5, 6]. The structure and distribution of the genes were also compared using *in silico* MolecularCloning ver. 7. The 3D-structure of FA desaturase gene products found in *Neptuniibacter* strains was predicted using Phyre2 ver. 2 [23] with the same approach described previously [5, 6].

## Results and discussion

### Molecular phylogenetic analysis based on 16S rRNA gene nucleotide sequences

Phylogenetic analysis based on 16S rRNA gene nucleotide sequences showed that the strain PT1[T] was affiliated to the members of the genus *Neptuniibacter* showing 96.7–98.2% sequence similarities, which are below the proposed threshold range for species boundary at 98.7% [24, 25]. The strain showed high sequence similarities of 98.2% with *N. marinus*. The tree also revealed that the PT1[T] formed a robust monophyletic cluster with *N. caesariensis*, *N. halophilus*, *N. marinus*, and *N. pectenicola* within the genus *Neptuniibacter* (Fig 1).

The ML tree was reconstructed using the K2+G+I model. Numbers shown on branches are bootstrap values (%) based on 1,000 replicates (>70%). A total of 1,284 bp was compared (162–1,446 position in PT1[T], GenBank accession number AP025763). 16S rRNA gene nucleotide sequences loci r00010 and r00040 under GenBank accession number GCF_000974885.1 were used as an outgroup to generate this rooted tree. Bar, 0.05 substitutions per nucleotide position.

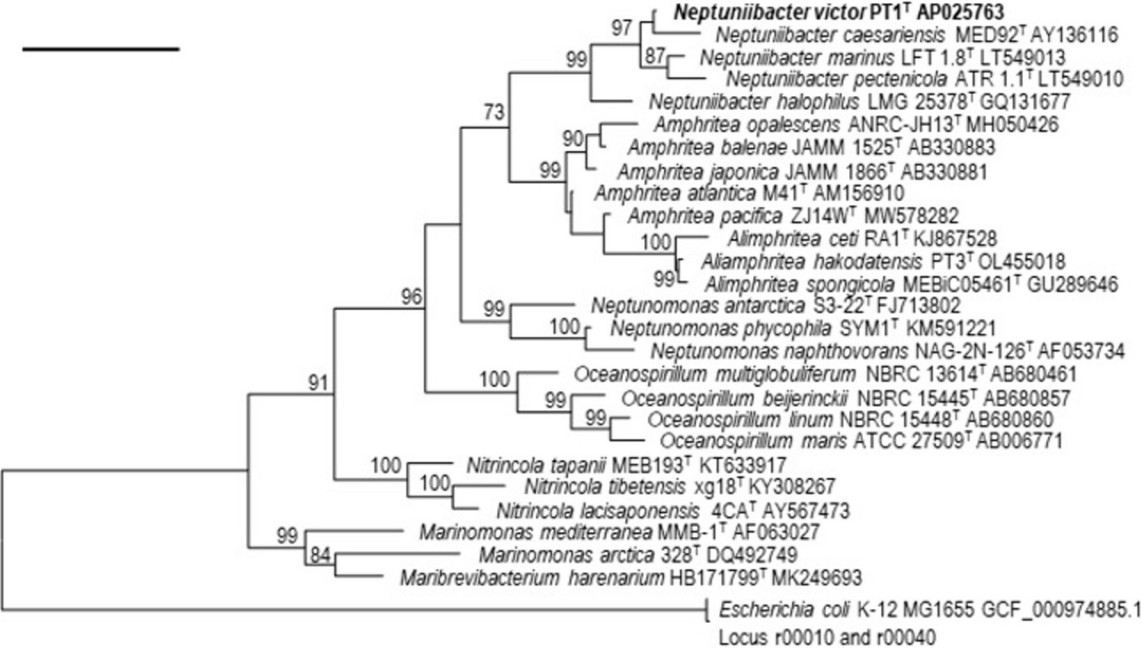

**Fig 1. ML tree based on 16S rRNA gene nucleotide sequences of strain PT1[T] and related type strains.**

## Genomic features and overall genome relatedness indices (OGRIs)

The ANI values of the PT1$^T$ against *N. halophilus*, *N. caesariensis*, *N. marinus*, and *N. pectenicola* were 72.0%, 72.4%, 74.8%, and 74.7%, respectively (S1 Fig), which are below the species boundary threshold of 95% proposed in previous studies [26]. The DDH values of PT1$^T$ against those species were 18.7%, 18.3%, 19.5%, and 19.1%, respectively, which were also below the species delineation threshold (70%). ANI and *in silico* DDH revealed PT1$^T$ as a novel species.

The AAI values of PT1$^T$ for against *N. halophilus*, *N. caesariensis*, *N. marinus*, and *N. pectenicola* were 75.8%, 74.8%, 77.8%, and 77.7%, respectively, which are below the 95% species boundary and above the 64–67% genus boundary, indicating that they are new species within the genus (S2 Fig) [5, 6, 8, 27].

## Multilocus sequence analysis (MLSA)

MLSA network showed that PT1$^T$ is likely to be monophyletic with previously described *Neptuniibacter* species but conspecifically separate from them. This result supports the proposal that PT1$^T$ is a novel species in the genus *Neptuniibacter* (Fig 2).

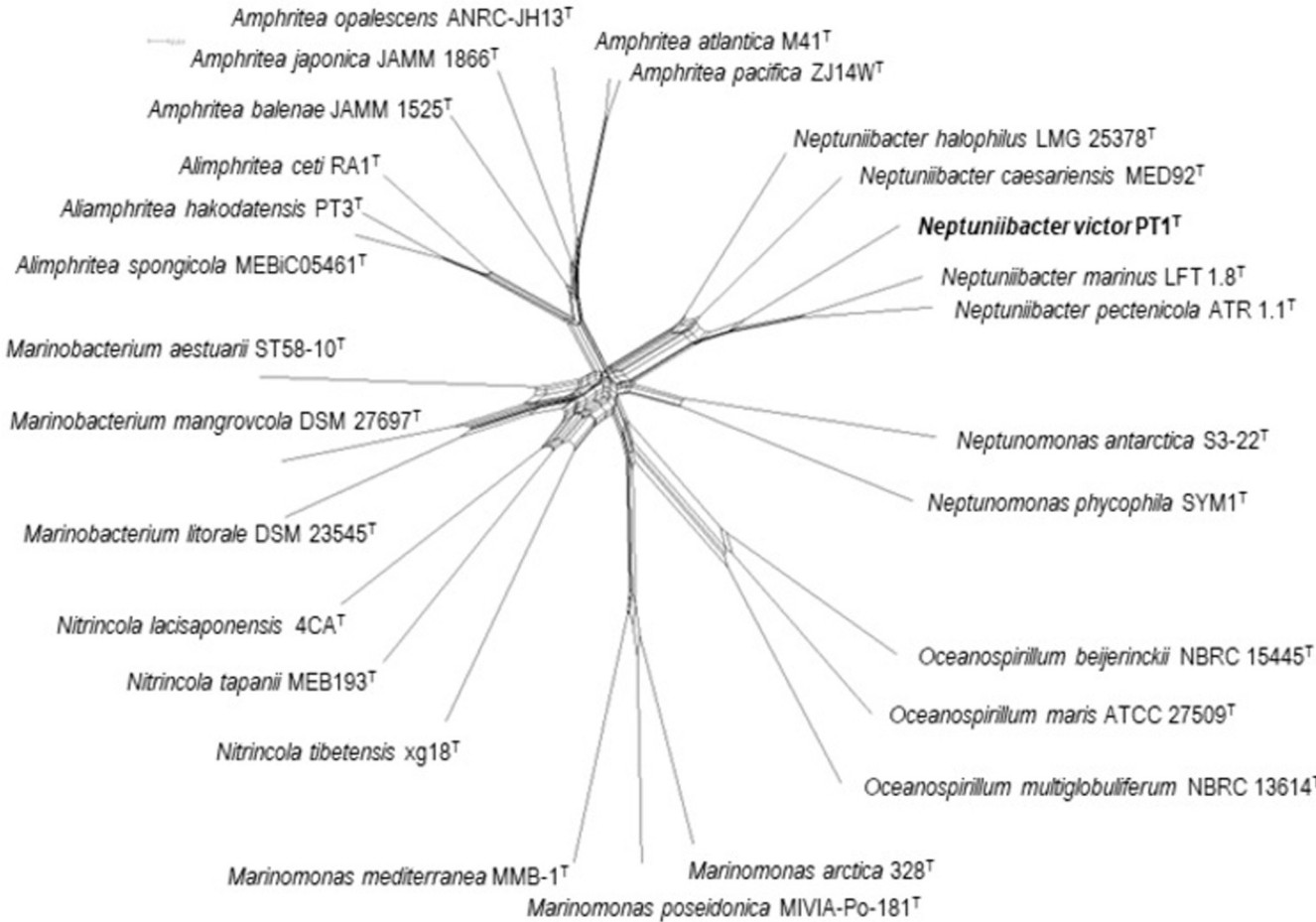

**Fig 2. MLSA network of PT1$^T$ and related *Oceanospirillaceae*.** A list of strains and their assembly accession is provided in Table 1. Sequence regions used for reconstructing the network were 1–995 on *mreB*, 1–1,041 on *recA*, 1–981 on *rpoA*, and 1–2,648 on *topA*. The clade which PT1$^T$ belonged was robust with supported by 100% bootstrap.

## Experimental phenotypic characterization and *in silico* phenotyping

Several experimental phenotypic characterization tests revealed that PT1[T] shares several characteristics with *Neptuniibacter* spp. including being positive for oxidase and growth at 15˚C and 30˚C [1–3]. PT1[T] could be differentiated from *Neptuniibacter* spp. by growth at 4˚C, 10˚C, and 37˚C, catalase reaction, hydrolysis of starch and DNA, nitrate reduction, and utilization of glucose, and DL-malic acid (Table 2).

Unfortunately, PT1[T] was unable to be grown in manually prepared media commonly used for OF, salt requirement, and carbon assimilation tests [5–8, 10], so we also use *in silico* phenotyping approach using Traitar software for predicting the phenotype. Before predicting PT1[T] phenotype, the accuracy of Traitar phenotyping was evaluated to be 81.5% in average using phenotypic characterization from four described *Neptuniibacter* species (*N. halophilus*, *N. caesariensis*, *N. marinus*, and *N. pectenicola*) with 40–47 traits compared. The Traitar prediction showed that PT1[T] was differentiated by nine traits among *Neptuniibacter* species: utilization of acetate (accuracy 50%), cellobiose (accuracy 100%), maltose (accuracy 75%) and mucate (not determined), hydrolysis of casein (accuracy 50%), production of alkaline phosphatase (accuracy 75%), indole (accuracy 75%), lysine decarboxylase (not determined), and growth at 42˚C (accuracy 25%) (S3 Fig). However, no maltose utilization and indole production were observed using the API20NE.

## Pan and core genomes and its ecogenomic interpretation

The pangenome of *Neptuniibacter* species consists of 6,605 gene clusters (17,748 genes) (Fig 3). Genes were classified into *Core*, present in all strains, and *Unique* in individual species. *Core* consisted of 1,973 gene clusters (10,215 genes). COG categories such as J (translation), K (transcription) and E (amino acid transport) were abundant in this bin. *Unique* in PT1[T] consists of 596 gene clusters (612 genes). COG categories such as E (amino acid transport system) and I (lipid transport and metabolism) were included in this bin. Genes related to nitrogen metabolism were also detected in *Unique* in PT1[T].

**Table 2. Phenotypic characteristics of PT1[T] and *Neptuniibacter* strains.**

| Characteristics | *N. victor* sp. nov. PT1[T] | *N. halophilus*[T] | *N. caesariensis*[T] | *N. marinus*[T] | *N. pectenicola*[T] |
|---|---|---|---|---|---|
| **Growth at** | | | | | |
| 4˚C | - | - | - | + | + |
| 10˚C | + | - | + | + | + |
| 37˚C | - | + | + | - | + |
| 45˚C | - | - | - | - | - |
| **Catalase** | - | - | + | - | + |
| **Production of** | | | | | |
| amylase | - | - | - | + | - |
| DNase | w | w | - | nd | nd |
| **Nitrate reduction** | + | - | - | - | - |
| **Utilization of** | | | | | |
| D-glucose | + | - | - | - | - |
| DL-malic acid | - | - | + | + | + |

Phenotypic data for *Neptuniibacter* are from this and previous studies [1–3]. All strains are positive in growth at 15–30˚C, oxidase test. All strains are negative in growth at 45˚C, indole production, utilization of L-arabinose, D-mannitol, N-acetyl-D-glucosamine, maltose, and hydrolysis of Tween 20 and 80, and gelatin.
+: present; -: lack; w: weak reaction; nd: not determined.

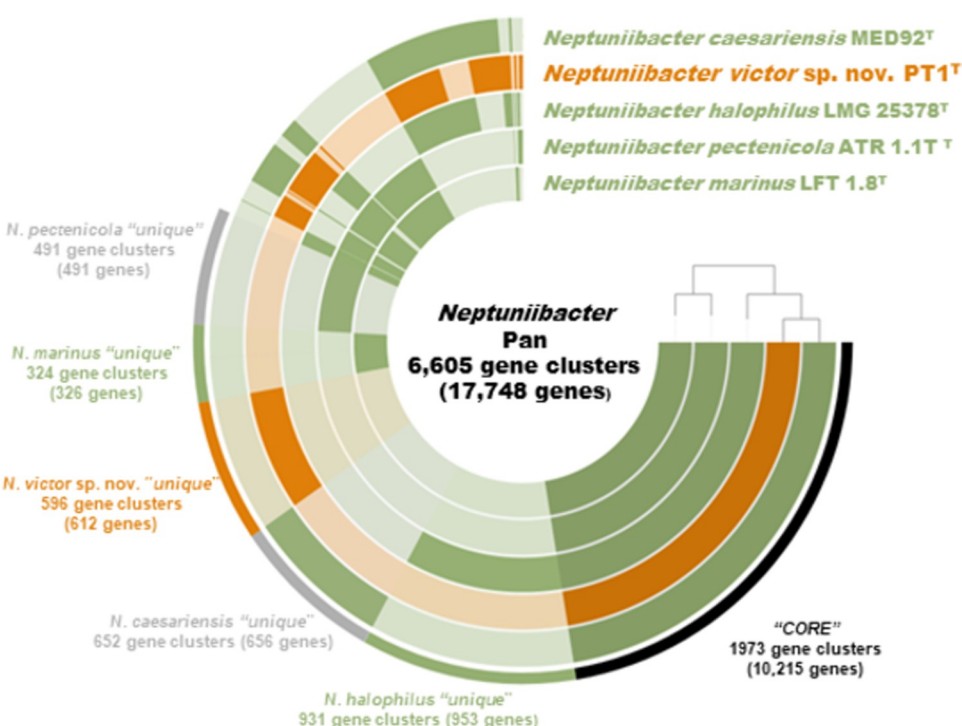

**Fig 3. Anvi'o representation of the pangenome of strain PT1[T] and *Neptuniibacter*.** Layers represent each genome, and the darker areas represent the occurrence of gene clusters. The outer colored bars indicate the "*Core*" or "*Unique*" bins.

Interestingly, PT1[T] possessed a 45 kb gene island encoding a type VI secretion system (T6SS) and genes (*narGHJI* and *norBC*) responsible for performing several steps of denitrification, nitrate and nitric oxide reductions. These genes were grouped into the PT1-*Unique* genes by pangenomic analysis. Recently, T6SS has been identified as having a role in killing phenotypes of *Vibrio fischeri* to establish spatial separation in different crypts of a light organ in the host *Euprymna scolopes* squid [28, 29]. NarGHI is typically identified as a dissimilatory nitrate reductase with a role in nitrate respiration under not only anaerobic but also aerobic conditions [30, 31]. The presence of formate dehydrogenase genes suggests that formate could be served as an electron donor for nitrate respiration in PT1[T] [30]. Nitrite is further metabolized by a nitrate reductase NirBD to ammonia in the PT1[T]. The assimilatory nitrate reduction pathway using NasAC and NirA has also been predicted in this strain [32]. Detoxification of nitric oxide (NO), which is a multifunctional immune trigger involving antimicrobial activity, using a nitric oxide reductase, NorBC, has been known as a function in sustaining host-microbes interactions in a wide variety of animals and plants [33]. As PT1 possesses similar sequences to those of the key abundant ASVs associated with the pentactula larvae of *A. japonicus*, the T6SS with nitrate respiration and NO detoxification genotypes of the PT1[T] could contribute to competitive dominance in the early life of the benthic host animal. Genes responsible for poly-β-hydroxybutyrate biosynthesis were found in the PT1[T] genome [34], but the effects of the PT1[T] on promoting growth in sea cucumber *A. japonicus* have yet to be observed (data not shown).

## *In silico* chemical taxonomy

As fatty acids, polar lipids, and isoprenoid quinones are common subjects for chemical taxonomic analyses, *in silico* chemical taxonomy among the *Neptuniibacter* species including the

newly described species of PT1$^T$ based on genome sequences, as an alternative to the traditional chemical taxonomy, was performed (S1 Table).

Comparative genome analyses among described *Neptuniibacter* species referring reported cellular fatty acids of *Neptuniibacter* species, which are mainly linear, mono-unsaturated or saturated consisting of C16:0, C16:1 and C18:1, with a small amount of C10:0 3-OH (S1 Table) [1–3], reconstructed the basic type II fatty acid biosynthesis (FAS II) pathway driven by *fabABFDGIVYZ*, which is very similar to that of *E. coli* [35] (S4–S6 Figs). In particular, long-chain saturated fatty acid (C16:0) is one of the main features of *Neptuniibacter* fatty acids, comprising approximately 15–26% of the total (S1 Table). C16:0 is one of the main products of the FAS II pathway, suggesting PT1$^T$ could also produce C16:0 based on genome comparisons (S6 Fig). In *Neptunibacter* species, C16:1 and C18:1 together make up over 60% of the total fatty acids (S1 Table), so monounsaturated fatty acids are also major features of the fatty acid profile in PT1$^T$. Monounsaturated fatty acids can be produced through ω7 monounsaturated fatty acid synthesis initiated by isomerization of trans-2-decenoyl-ACP into cis-3-decenoyl-ACP by *fabA* gene product (S4 Fig). After elongation by *fabB* gene product, the acyl chain is returned to the FAS II pathway and goes through further elongation, producing C16:1ω7c and C18:1ω7c [5, 6, 36]. The genome comparisons predicted that all strains including PT1$^T$ are capable of producing C16:1ω7c and C18:1ω7c by *fabA* and *fabB* gene products. 3-hydroxylated FAs, which are the primary fatty acids in lipid A as well as in ornithine-containing lipids, could be supplied by the FAS II pathway since 3-hydroxy-acyl-ACP is known to be normally intermediated in the FAS II elongation cycle [5, 6, 36]. Since no genes responsible for the synthesis of ornithine-containing lipids were found in genomes of PT1$^T$ or any other species in either genus, it is likely that 3-hydroxylated FAs in PT1$^T$ originate in lipid A [5, 6].

A comparative genome survey of the genes responsible for the FAS II pathway on the PT1$^T$ genome reveals the presence of a core gene set, and genomic structures of FAS II core genes are likely to be retained between described *Neptuniibacter* species (S5 and S6 Figs), which could lead to the conclusion that the novel strain is capable of producing similar FA profiles, mainly consisting of C16:0, C16:1ω7c, and C18:1ω7c. PT1$^T$ is also potentially capable of making C10:0 3-OH which is commonly found in this genus because of the presence of the *lpxA* gene. *lpxA* is responsible for the incorporation of 3-hydroxyacyl to UDP-N-acetyl-α-D-glucosamine, which is a primary reaction to the biosynthesis of lipid A [37].

Comparative analysis among *Neptuniibacter* species including PT1$^T$ also revealed a complete gene set for the production of PG and PE; *plsX*, *plsY*, *plsC*, *cdsA*, *pssA*, *psd*, *pgsA* and *pgpA* (S5 and S6 Figs), which indicate that PT1$^T$ produces PG and PE as major polar lipids, similar to other *Neptuniibacter* species.

The only respiratory quinone reported from previously described *Neptuniibacter* species is ubiquinone-8 (Q-8) (S1 Table). Biosynthesis of Q-8 consists of nine steps, and Ubi proteins are involved in each reaction; side chain synthesis by *ispB* gene product, core biosynthetic pathway by *ubiC*, *ubiA*, *ubiD*, *ubiX*, *ubiI*, *ubiG*, *ubiH*, *ubiE*, and *ubiF*, and accessory hypothetical functions by three genes *ubiB*, *ubiJ*, and *ubiK* (S7 Fig) [5, 6, 38]. The set of genes was identified in PT1$^T$ genome, of which results strongly suggest that the predominant ubiquinone of PT1$^T$ is Q-8 (S7 Fig).

## Conclusions

A combination of modern genome taxonomic studies including *in silico* chemical taxonomy revealed that strain PT1$^T$ is a new species in the genus *Neptuniibacter*. The name *Neptuniibacter victor* sp. nov. (PT1$^T$ = JCM 35563$^T$ = LMG 32868$^T$) is proposed.

## Description of *Neptuniibacter victor* sp. nov.

*Neptuniibacter victor* (vic'tor. L. masc. n. *victor*, the winner, referring to the predicted ecophysiology of this bacterium possessing T6SS in sea cucumber aquaria, where the bacterium was isolated).

Gram-negative, motile and aerobic rod. Colonies on Marine Agar (BD) are white and 1.0–3.0 mm in diameter after culture for 3 days. No pigmentation and bioluminescence are observed. Growth occurs at 15˚C-30˚C. Positive for oxidase. Weakly positive for DNase. Negative for growth on a manually prepare basal seawater medium, catalase, hydrolyses of starch, agar, Tween 20 and 80, and gelatin. The newly described species is characterized by positive for nitrate to nitrite, and utilization of D-glucose, using API20NE. The DNA G+C content is 44.2% and the genome size is 3.95 Mb.

The type strain PT1$^T$ (= JCM 35563$^T$ = LMG 32868$^T$) was isolated from a pentactula larvae of *A. japonicus* reared in a laboratory aquarium at Hokkaido University, Hakodate, Japan. The GenBank accession number for the 16S rRNA gene sequence of the type strain is LC716006. The complete genome sequence of the strain is deposited in the DDBJ/ENA/GenBank under the accession number AP025763.

## Supporting information

**S1 Table. Chemotaxonomic profile of previously described *Neptuniibacter*.** PG, phosphatidylglycerol; PE, phosphatidylethanolamine; DPG, diphosphatidylglycerol; PL, phospholipids; AL, aminolipid; PN, phosphoaminolipid; nd: not determined.
(PDF)

**S1 Fig. Heat map representation of ANI values.**
(PDF)

**S2 Fig. Heat map representation of AAI values.** Reference genomes were retrieved from NCBI database.
(PDF)

**S3 Fig. *In silico* phenotyping of PT1 by Traitar.** 0: negative, 1:phypat positive, 2: phypat +PGL positive, 3: both predictions positive.
(PDF)

**S4 Fig. Predicted fatty acid synthetic pathway in *Neptuniibacter*.** Predicted fatty acid synthetic pathway in *Neptuniibacter*. ACP: acyl-carrier protein; *accABCD*: acetyl-CoA carboxylase complex; *fabA*: 3-hydroxyacyl-ACP dehydrase/trans-2-decanoyl-ACP isomerase; *fabB*: 3-ketoacyl-ACP synthase I; *fabD*: malonyl-CoA: ACP transacylase; *fabF*: 3-ketoacyl-ACP synthase II; *fabG*: 3-ketoacyl-ACP reductase; *fabI*: enoyl-ACP reductase I; *fabV*: enoyl-ACP reductase; *fabY*: 3-ketoacyl-ACP synthase; *fabZ*: 3-hydroxyacyl-ACP dehydratase; *lpxA*: glucosamine N-acyltransferase.
(PDF)

**S5 Fig. Gene structure of FAS associated genes.**
(PDF)

**S6 Fig. Genomic distribution of *fab* and associated genes.** Protein/enzyme name each gene is coding: *fabA*: 3-hydroxyacyl-ACP dehydrase/trans-2-decanoyl-ACP isomerase; *fabB*: 3-ketoacyl-ACP synthase I; *fabD*: malonyl-CoA: ACP transacylase; *fabF*: 3-ketoacyl-ACP synthase II; *fabG*: 3-ketoacyl-ACP reductase; *fabI*: enoyl-ACP reductase I; *fabV*: enoyl-ACP reductase; *fabY*: 3-ketoacyl-ACP synthase; *fabZ*: 3-hydroxyacyl-ACP dehydratase; *acpP*: Acyl-

carrier protein; *plsX*: FA/phospholipid synthesis protein; *lpxAD*: glucosamine N-acyltransferase; *lpxB*: lipid-A-disaccharide synthase.
(PDF)

**S7 Fig. Predicted Q-8 synthetic pathways in *Neptuniibacter*.**
(PDF)

## Acknowledgments

We gratefully thank for Professor (Emeritus) Aharon Oren, The Hebrew University of Jerusalem, for his advice on bacterial names.

## Author Contributions

**Conceptualization:** Juanwen Yu, Sayaka Mino, Yuichi Sakai, Tomoo Sawabe.

**Data curation:** Rika Kudo, Ryota Yamano, Juanwen Yu, Shotaro Koike, Jiro Tsuchiya, Sayaka Mino, Tomoo Sawabe.

**Formal analysis:** Rika Kudo, Ryota Yamano, Juanwen Yu, Alfabetian Harjuno Condro Haditomo, Mayanne A. M. de Freitas, Tomoo Sawabe.

**Funding acquisition:** Sayaka Mino, Tomoo Sawabe.

**Investigation:** Yuichi Sakai, Tomoo Sawabe.

**Methodology:** Rika Kudo, Ryota Yamano, Shotaro Koike, Alfabetian Harjuno Condro Haditomo, Mayanne A. M. de Freitas, Jiro Tsuchiya, Sayaka Mino, Tomoo Sawabe.

**Project administration:** Juanwen Yu, Tomoo Sawabe.

**Resources:** Jesús L. Romalde, Tomoo Sawabe.

**Supervision:** Sayaka Mino, Yuichi Sakai, Tomoo Sawabe.

**Writing – original draft:** Rika Kudo, Ryota Yamano, Juanwen Yu, Shotaro Koike, Mayanne A. M. de Freitas, Tomoo Sawabe.

**Writing – review & editing:** Rika Kudo, Ryota Yamano, Juanwen Yu, Shotaro Koike, Alfabetian Harjuno Condro Haditomo, Mayanne A. M. de Freitas, Jiro Tsuchiya, Sayaka Mino, Fabiano Thompson, Jesús L. Romalde, Hisae Kasai, Yuichi Sakai, Tomoo Sawabe.

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
