## [Editor Report · Decision Letter 0]

7 Mar 2023

PONE-D-23-05318Genome taxonomy of the genus Neptuniibacter and proposal of Neptuniibacter victor sp. nov. isolated from sea cucumber larvae.PLOS ONE

Dear Dr. Sawabe,

Thank you for submitting your manuscript to PLOS ONE. Before I'll send your manuscript to reviewers, I need to secure that the manuscript meets publication criteria. Since the genome assembly is a part of your manusript for the sake of reproducibility you need to make all data publicly available. Therefore, you need to upload raw sequencing data to Sequence Read Archive or similar database. Moreover, your accessions for *N. halophilus* LMG 25378T are not currently available in GenBank/EMBL/DDBJ. Please make all neccessary data available for peer review.

We look forward to receiving your revised manuscript.

Kind regards,

Karel Sedlar, Ph.D.

Academic Editor

PLOS ONE
---

## [Author Response · Author response to Decision Letter 0]

13 Mar 2023

Dear Dr. Sedlar, Academic Editor, PLOS ONE

We gratefully thank to the editor for comments concerning our manuscript entitled “Genome taxonomy of the genus Neptuniibacter and proposal of Neptuniibacter victor sp. nov. isolated from sea cucumber larvae”, PONE-D-23-05318.

We have described accession numbers and these are now publicly available.

Academic Editor: Thank you for submitting your manuscript to PLOS ONE. Before I'll send your manuscript to reviewers, I need to secure that the manuscript meets publication criteria. Since the genome assembly is a part of your manuscript for the sake of reproducibility you need to make all data publicly available. Therefore, you need to upload raw sequencing data to Sequence Read Archive or similar database. Moreover, your accessions for N. halophilus LMG 25378T are not currently available in GenBank/EMBL/DDBJ. Please make all necessary data available for peer review.

Reply: Thank you for your comments. We provide DRA numbers and all sequences are now publicly available.

---

## [Decision Letter · Decision Letter 1]

6 Jun 2023

PONE-D-23-05318R1

Genome taxonomy of the genus Neptuniibacter and proposal of Neptuniibacter victor sp. nov. isolated from sea cucumber larvae.

PLOS ONE

Dear Dr. Sawabe,

Thank you for submitting your manuscript to PLOS ONE. After careful consideration, we feel that it has merit but does not fully meet PLOS ONE’s publication criteria as it currently stands. Therefore, we invite you to submit a revised version of the manuscript that addresses the points raised during the review process.

Please try to address all reviewers’ comments, even though they both proposed only a minor revision. Although the reproducibility of your study was made possible by making you data publicly available, there are still some concerns raised by the first reviewer. Please make sure the methodology is written into detail, including description of particular parameters used for computations and versions of used computational tools. I think it would be much appreciated if you consider reviewer 2’s comment on performing additional in vitro testing of phenotypic traits using standardized commercial kit as this is a good practice when proposing a novel species. This would also help you to gather additional information utilizable for BacDive database which would make your bacterium more visible to scientific community and could bring more readers to your paper. If you think these experiments are replaceable by computational analysis, please try to explain why you think so.

We look forward to receiving your revised manuscript.

Kind regards,

Karel Sedlar, Ph.D.

Academic Editor

PLOS ONE

Journal Requirements:

Additional Staff Editor Comments:

During our internal evaluation of the manuscript, we found significant text overlap between your submission and your previously published work (In silico chemical taxonomy section in particular):

https://journals.plos.org/plosone/article?id=10.1371%2Fjournal.pone.0271174

Please revise the manuscript to rephrase the duplicated text and fully cite all your sources, where appropriate.

Reviewers' comments:

Reviewer's Responses to Questions

**Comments to the Author**

1. If the authors have adequately addressed your comments raised in a previous round of review and you feel that this manuscript is now acceptable for publication, you may indicate that here to bypass the “Comments to the Author” section, enter your conflict of interest statement in the “Confidential to Editor” section, and submit your "Accept" recommendation.

Reviewer #1: (No Response)

Reviewer #2: (No Response)

2. Is the manuscript technically sound, and do the data support the conclusions?

Reviewer #1: Yes

Reviewer #2: Yes

3. Has the statistical analysis been performed appropriately and rigorously? 

Reviewer #1: Yes

Reviewer #2: I Don't Know

4. Have the authors made all data underlying the findings in their manuscript fully available?

Reviewer #1: Yes

Reviewer #2: Yes

5. Is the manuscript presented in an intelligible fashion and written in standard English?

Reviewer #1: Yes

Reviewer #2: Yes

6. Review Comments to the Author

Reviewer #1: The article "Genome taxonomy of the genus Neptuniibacter and proposal of Neptuniibacter victor. sp nov. isolated from sea cucumber larvae" provides insights into a newly discovered strain PT1T found in sea cucumber larvae where genotypic and phenotypic in silico analysis is conducted. The article is well-written and provides a clear description of the research findings. However, several points need to be clarified.

Major comments:

The article mentions that the Unicycler tool was used for genome assembly. However, it is not clear if any preprocessing was done on the Illumina data, such as adapter removal, which can significantly influence results. Moreover, I am missing information on whether the quality of the sequencing data was checked.

The author states that whole genome sequences were obtained, and as a result, contigs were obtained (Tab. 1). In the case of more contigs, it is unclear if contigs contain only chromosome sequences or if plasmids are also present in the assembled contigs. If yes, are genes from plasmids included in the pan-genome analysis?

On line 107, the article states that the ML tree was generated from 100 replicates, but Figure 1 indicates that 1000 replicates were performed. This discrepancy should be clarified.

The article references the acceptance of results according to formula 2 for the OGRIs calculation (line 115), but formula 2 is not provided.

While AAI values were calculated for PT1T and other Neptuniibacter species, the obtained values are not provided. It is unclear how the author determined which genomes were used or not for analysis, as indicated in Table 1. The same issue applies to the MLSA column in Table 1.

The article states that phylogenetic network reconstruction was performed for MLSA, but there is no description of the settings used for the SplitsTree tool or the method of construction used.

In the section on pan and core genome analysis, it is not clear if the same tool was used for both analyses.

In the section Pan and core genomes and its ecogenomic interpretation, the author mentions that the assimilatory nitrate reduction pathway was predicted in the analyzed strain. It is unclear how this pathway was predicted and if a specific tool was used.

There are some minor comments regarding the article that I would like to make:

The version number should be included for each tool used in this study (e.g. lines 74, 113, 148 etc.).

The citation for the MEGAX tool is missing (line 105).

The word "Illumina" should start with a capital letter (line 91).

In Tab. 1, there should be some units for the size column.

Citations need to be sorted in ascending order (line 172).

The abbreviation OF is used, but its meaning is not explained (line 72).

Reviewer #2: PONE-D-23-05318R1

The authors present a study: Genome taxonomy of the genus Neptuniibacter and a proposal of Neptuniibacter victor sp. nov. isolated from sea cucumber larvae.

The methods used demonstrably confirm that the described strain PT1T belongs to a new, previously undescribed Neptuniibacter species. The article is supported by sufficient literature references.

The sequencing methods, genome analysis and MLST analysis are described and performed in detail and provide valuable information. On the other hand, my main complaint is that the article provides chemotaxonomic results and phenotyping mainly from the in silico approach. It brings mainly predicted phenotypic characteristics of the studied strain PT1T. I miss in the discussion any mention of the reliability of the analysis with Traitar (Microbial Trait Analyzer) and the used in silico chemical taxonomy and maybe comparison with in vitro methods. In my opinion, a description of genomic and phenotypic traits (tested in vitro) should be more balanced.

I recommend the article for publication with minor revision.

Other comments and recommendations:

Line 127, Table 1: It would be useful to adjust the width of the columns to make the data more readable.

Line 185, Table 2: The numbers of the reference type strains should be indicated.

Table 2: The range of the temperatures tested is too wide (4°C, 10°C, ...37°C). It seems reasonable and logical to test the growth also at 20 or 25°C.

The phenotyping data are rather insufficient for the description of the novel species. I suggest testing the commercial kit API ZYM to complement the phenotypic description. This commercial kit is well standardised and has been used in a species description of other Neptuniibacter spp. Also test a hydrolysis of gelatine and Tween 80 if possible (tests mentioned in previous Neptuniibacter species).

Line 187: Please complete the following sentence, there is something missing in the text: Phenotypic data for Neptuniibacter .... are from this study and .... [1-3].

Line 207: Please complete the following sentence, there is something missing in the text: Fig 3. Anvi'o representation of the pangenome of strain PT1T and Neptuniibacter ...

The word Unique, written in italics, is somewhat confusing together with the species name in Fig. 3. It could be written in brackets.

Please include the reference strain numbers in all tables and figures; both in the main text and in the supplementary data.

Line 310: There is a mistake in the doi number. The correct one is doi:310 10.1099/ijs.0.64524-0.

Please add to the discussion section information on the reliability/confidence of the in silico approaches and assess their suitability for the genera studied.

It would be interesting to compare the predicted properties (Fig. S3) with in vitro tests from previous descriptions of Neptuniibacter spp. Do the results agree? In which cases is the gene not expressed?

Fig S7: There is an extra dot.

7. PLOS authors have the option to publish the peer review history of their article (what does this mean?). If published, this will include your full peer review and any attached files.

Reviewer #1: **Yes: **Marketa Nykrynova

Reviewer #2: No

---

## [Author Response · Author response to Decision Letter 1]

1 Jul 2023

Dear Professor Karel Sedlar, the editor, and reviewers,

We appreciate editor and reviewers for constructive suggestions for PONE-D-23-05318R1. We improved the manuscript according to the editorial office and reviewers’ comments. Responses for specific comments are described as follows. All changes were found in the tracking-on file separately uploaded. We also refine Tables 1 and 2, and Figs 1-3. Table 1 was not framed in body, so we also attached table1.tif as other material.

Editorial office

1. During our internal evaluation of the manuscript, we found significant text overlap between your submission and your previously published work (In silico chemical taxonomy section in particular)

Response: Thank you for the suggestion. We used almost same approach of previously published papers, so many parts of body text were overlapped. However, we did remove redundancy and put citations of previous paper.

Reviewer: 1

1. The article mentions that the Unicycler tool was used for genome assembly. However, it is not clear if any preprocessing was done on the Illumina data, such as adapter removal, which can significantly influence results. Moreover, I am missing information on whether the quality of the sequencing data was checked.

Response: Thank you so much for the comments after careful reading. We did not perform preprocessing Illumina reads at this time because we confirmed QV>27 and no adaptor and barcode sequences of the Illumina read using the FastQC program before the assembly. We put the explanation in body.

2. For pangenome analyses;

1) The author states that whole genome sequences were obtained, and as a result, contigs were obtained (Tab. 1). In the case of more contigs, it is unclear if contigs contain only chromosome sequences or if plasmids are also present in the assembled contigs. If yes, are genes from plasmids included in the pan-genome analysis? 

2) In the section on pan and core genome analysis, it is not clear if the same tool was used for both analyses.

3) In the section Pan and core genomes and its ecogenomic interpretation, the author mentions that the assimilatory nitrate reduction pathway was predicted in the analyzed strain. It is unclear how this pathway was predicted and if a specific tool was used.

Response: 

1) Thank you so much for the comments. Actually, pan-genome analyses only use genomes, but unfortunately, we did not obtain complete genome sequences ourselves from the other three Neptuniibacter type strains. Due to being not survived during shipment from European collections by the heat wave in Europe in 2022 (too hot summer during shipments), we were unable to obtain those type strains. So, we decided to use draft genomes obtained from public databases. These sequences might include plasmids, but we could not select and differentiate them. PT1 has no plasmid, but N. halophilus has one plasmid. 

2) We used Anvio for both pan- and core- analyses. We modified body in related parts.

3) For assimilatory nitrate reduction, we evaluated only by presence of nasAC and nirA genes unique in PT1. We added one more reference for that. 

3. For the other methodology

1) On line 107, the article states that the ML tree was generated from 100 replicates, but Figure 1 indicates that 1000 replicates were performed. This discrepancy should be clarified. 

2) The article references the acceptance of results according to formula 2 for the OGRIs calculation (line 115), but formula 2 is not provided. 

3) While AAI values were calculated for PT1T and other Neptuniibacter species, the obtained values are not provided. 

4) It is unclear how the author determined which genomes were used or not for analysis, as indicated in Table 1. The same issue applies to the MLSA column in Table 1.

5) The article states that phylogenetic network reconstruction was performed for MLSA, but there is no description of the settings used for the SplitsTree tool or the method of construction used.

Response: Thank you for your comments. All points were fixed and/or corrected accordingly. Rerated to 1) and 5), we refined 16S and MLSA network as well, in particular, regions used for those phylogenetic analyses were described. For 2), we modified the body to cover formula 2 explanation by ref 17, in which it is explained that Formula 2 is optimized for using incomplete genomes. We both analyzed complete and incomplete genomes, so we selected formula 2. AAI values were already provided in body around L170 and S2 Fig, but improved the body. To be clear which genome we determined ourselves, we put “in this study” in Table 1. For MLSA, “concatenation of sequences and phylogenetic NeighborNet reconstruction were performed using SplitsTree 4.16.2 with options of Jukes-Cantor correction, gap-exclusion and 1,000 bootstrap”, but eliminating redundancy of previously published paper (Yamano et al., 2023, PLoS One), we add this ref.

4. There are some minor comments regarding the article that I would like to make:

1) The version number should be included for each tool used in this study (e.g. lines 74, 113, 148 etc.).

2) The citation for the MEGAX tool is missing (line 105).

3) The word "Illumina" should start with a capital letter (line 91).

4) In Tab. 1, there should be some units for the size column.

5) Citations need to be sorted in ascending order (line 172).

6) The abbreviation OF is used, but its meaning is not explained (line 72).

Response: Thank you for your comments. All points were fixed and/or corrected accordingly.

Reviewer: 2 

1. The authors present a study: Genome taxonomy of the genus Neptuniibacter and a proposal of Neptuniibacter victor sp. nov. isolated from sea cucumber larvae. The methods used demonstrably confirm that the described strain PT1T belongs to a new, previously undescribed Neptuniibacter species. The article is supported by sufficient literature references. The sequencing methods, genome analysis and MLST analysis are described and performed in detail and provide valuable information. On the other hand, my main complaint is that the article provides chemotaxonomic results and phenotyping mainly from the in silico approach. It brings mainly predicted phenotypic characteristics of the studied strain PT1T. I miss in the discussion any mention of the reliability of the analysis with Traitar (Microbial Trait Analyzer) and the used in silico chemical taxonomy and maybe comparison with in vitro methods. In my opinion, a description of genomic and phenotypic traits (tested in vitro) should be more balanced.

2. Other comments and recommendations:

1) Line 127, Table 1: It would be useful to adjust the width of the columns to make the data more readable.

2) Line 185, Table 2: The numbers of the reference type strains should be indicated.

Response: Corrected accordingly.

3) Table 2: The range of the temperatures tested is too wide (4°C, 10°C, ...37°C). It seems reasonable and logical to test the growth also at 20 or 25°C.

Response: We tested 4, 10, 15, 20, 25, 30, 37°C. As indicated in footnote, all strains grow at 15-30°C.

4) The phenotyping data are rather insufficient for the description of the novel species. I suggest testing the commercial kit API ZYM to complement the phenotypic description. This commercial kit is well standardised and has been used in a species description of other Neptuniibacter spp. Also test a hydrolysis of gelatine and Tween 80 if possible (tests mentioned in previous Neptuniibacter species).

Response: According to the comments, we did perform API20NE, but not APIzym due to no distribution currently in Japan. Tween 80 test was also performed, and described the negative results in body. Traitar prediction results were removed from the description section.

5) Line 187: Please complete the following sentence, there is something missing in the text: Phenotypic data for Neptuniibacter .... are from this study and .... [1-3].

Response: Corrected accordingly.

6) Line 207: Please complete the following sentence, there is something missing in the text: Fig 3. Anvi'o representation of the pangenome of strain PT1T and Neptuniibacter ... The word Unique, written in italics, is somewhat confusing together with the species name in Fig. 3. It could be written in brackets.

Response: Changed accordingly.

7) Please include the reference strain numbers in all tables and figures; both in the main text and in the supplementary data.

8) Line 310: There is a mistake in the doi number. The correct one is doi:310 10.1099/ijs.0.64524-0.

Response: Changed accordingly.

9) Please add to the discussion section information on the reliability/confidence of the in silico approaches and assess their suitability for the genera studied.

It would be interesting to compare the predicted properties (Fig. S3) with in vitro tests from previous descriptions of Neptuniibacter spp. Do the results agree? In which cases is the gene not expressed?

Response: Thank you so much for the comments. We did compared results by Traitar and experiments, and 82% accuracy in average was evaluated. We did add some results and sentences in related section. We also attached this result here (please see attached file separately). Finally, we did observe some growth in API20NE system, we did remove the Traitar prediction results from the description.

Fig S7: There is an extra dot.

Response: Changed accordingly.

---

## [Decision Letter · Decision Letter 2]

24 Jul 2023

PONE-D-23-05318R2Genome taxonomy of the genus Neptuniibacter and proposal of Neptuniibacter victor sp. nov. isolated from sea cucumber larvae.PLOS ONE

Dear Dr. Sawabe,

Thank you for submitting your manuscript to PLOS ONE. After careful consideration, we feel that it has merit but does not fully meet PLOS ONE’s publication criteria as it currently stands. Therefore, we invite you to submit a revised version of the manuscript that addresses the points raised during the review process.

 While I am overall happy with changes you made and I really appreciate you did additional experiment, reviewer 1 still found several issues with your manuscript. These are mostly grammar mistakes and small technical issues. Nevertheless, PLOS ONE is a respected journal which guarantees high technical quality of published articles. Therefore, I am forced to ask you for an additional round of revisions. Please be sure you adressed all issues that reviewer had raised and re-read your manuscript carefully to prevent yourselves from introducing any other mistakes. I believe you can address these issues in a very short time so your manuscript could be finally accepted for publication. Please submit your revised manuscript by Sep 04 2023 11:59PM. If you will need more time than this to complete your revisions, please reply to this message or contact the journal office at plosone@plos.org. Please include the following items when submitting your revised manuscript:A rebuttal letter that responds to each point raised by the academic editor and reviewer(s). You should upload this letter as a separate file labeled 'Response to Reviewers'.A marked-up copy of your manuscript that highlights changes made to the original version. You should upload this as a separate file labeled 'Revised Manuscript with Track Changes'.An unmarked version of your revised paper without tracked changes. You should upload this as a separate file labeled 'Manuscript'.If applicable, we recommend that you deposit your laboratory protocols in protocols.io to enhance the reproducibility of your results. Protocols.io assigns your protocol its own identifier (DOI) so that it can be cited independently in the future. For instructions see: https://journals.plos.org/plosone/s/submission-guidelines#loc-laboratory-protocols. Additionally, PLOS ONE offers an option for publishing peer-reviewed Lab Protocol articles, which describe protocols hosted on protocols.io. Read more information on sharing protocols at https://plos.org/protocols?utm_medium=editorial-email&utm_source=authorletters&utm_campaign=protocols.

We look forward to receiving your revised manuscript.

Kind regards,

Karel Sedlar, Ph.D.

Academic Editor

PLOS ONE

Journal Requirements:

Additional Staff Editor Comments:

Please address the text overlap concern we identified between your submission and your previously published work (In silico chemical taxonomy in the Result section in particular):

https://journals.plos.org/plosone/article?id=10.1371%2Fjournal.pone.0271174

Please revise the manuscript to rephrase the duplicated text and fully cite all your sources, where appropriate.

Reviewers' comments:

Reviewer's Responses to Questions

**Comments to the Author**

1. If the authors have adequately addressed your comments raised in a previous round of review and you feel that this manuscript is now acceptable for publication, you may indicate that here to bypass the “Comments to the Author” section, enter your conflict of interest statement in the “Confidential to Editor” section, and submit your "Accept" recommendation.

Reviewer #1: (No Response)

2. Is the manuscript technically sound, and do the data support the conclusions?

Reviewer #1: Yes

3. Has the statistical analysis been performed appropriately and rigorously? 

Reviewer #1: Yes

4. Have the authors made all data underlying the findings in their manuscript fully available?

Reviewer #1: Yes

5. Is the manuscript presented in an intelligible fashion and written in standard English?

Reviewer #1: Yes

6. Review Comments to the Author

Reviewer #1: For molecular phylogenetic analysis:

The authors added information about the settings used for MLSA (line 125). However, on line 131, they mentioned that "almost same setting" was used. I am missing information on how the setting was modified.

Other comments:

1) Line 28: There is an unnecessary gap between the range of numbers (72.8%- 74.8%).

2) Line 70: In the reference to citations, there are 2 hyphens instead of one.

3) Line 94: I guess QV means quality value, but in the text, it is written as quality vale. The word "Illumina" is missing the letter "m".

4) Line 98: There should be "assembly" instead of "assemble".

5) Line 120: The shortcut DDH is not explained.

6) Line 142: In the word "N. 10aesariensis" the number 10 should be removed.

7) Line 174: There should be "were" instead of "was".

8) Regarding the tools, the referencing of versions should be standardized. Sometimes it is written as "ver.," and other times as "v."

7. PLOS authors have the option to publish the peer review history of their article (what does this mean?). If published, this will include your full peer review and any attached files.

Reviewer #1: **Yes: **Markéta Nykrýnová

---

## [Author Response · Author response to Decision Letter 2]

25 Jul 2023

Dear Professor Karel Sedlar, the editor, and reviewers,

We appreciate editor and reviewers for constructive suggestions for PONE-D-23-05318R2. We improved the manuscript according to the editorial office and reviewers’ comments. Responses for specific comments are described as follows. All changes were found in the tracking-on file separately uploaded. Again, Table 1 was not framed in body, so we also attached table1.tif as other material.

Editorial office

1. Please address the text overlap concern we identified between your submission and your previously published work (In silico chemical taxonomy in the Result section in particular):

https://journals.plos.org/plosone/article?id=10.1371%2Fjournal.pone.0271174. We would like to make you aware that copying extracts from previous publications, especially outside the methods section, word-for-word is unacceptable. In addition, the reproduction of text from published reports has implications for the copyright that may apply to the publications. Please revise the manuscript to rephrase the duplicated text and fully cite all your sources, where appropriate.

Response: Thank you for the suggestion. Once again, we used almost same approach of previously published papers, so many parts of body text were overlapped. However, we did rewrite, remove redundancy, and add reference citations. Reference format was corrected accordingly.

Reviewer: 1

1. For molecular phylogenetic analysis:

The authors added information about the settings used for MLSA (line 125). However, on line 131, they mentioned that "almost same setting" was used. I am missing information on how the setting was modified.

Response: Thank you so much for the comments. Related part was improved, in particular, line 125 is fused to the first sentence of this section in removing redundancy.

Other comments:

1) Line 28: There is an unnecessary gap between the range of numbers (72.8%- 74.8%).

2) Line 70: In the reference to citations, there are 2 hyphens instead of one.

3) Line 94: I guess QV means quality value, but in the text, it is written as quality vale. The word "Illumina" is missing the letter "m".

4) Line 98: There should be "assembly" instead of "assemble".

5) Line 120: The shortcut DDH is not explained.

6) Line 142: In the word "N. 10aesariensis" the number 10 should be removed.

7) Line 174: There should be "were" instead of "was".

8) Regarding the tools, the referencing of versions should be standardized. Sometimes it is written as "ver.," and other times as "v."

Response: Thank you for the suggestion. All corrected accordingly.

---

## [Editor Report · Decision Letter 3]

2 Aug 2023

Genome taxonomy of the genus Neptuniibacter and proposal of Neptuniibacter victor sp. nov. isolated from sea cucumber larvae.

PONE-D-23-05318R3

Dear Dr. Sawabe,

We’re pleased to inform you that your manuscript has been judged scientifically suitable for publication and will be formally accepted for publication once it meets all outstanding technical requirements.

Kind regards,

Karel Sedlar, Ph.D.

Academic Editor

PLOS ONE

Additional Staff Editor Comments:

Please address the text overlap concern we identified between your submission and your previously published work (In silico chemical taxonomy in the Result section, in particular lines 259-290):

https://journals.plos.org/plosone/article?id=10.1371%2Fjournal.pone.0271174
---

## [Editor Report · Acceptance letter]

4 Aug 2023

PONE-D-23-05318R3 

Genome taxonomy of the genus *Neptuniibacter* and proposal of *Neptuniibacter victor *sp. nov. isolated from sea cucumber larvae 

Dear Dr. Sawabe:

I'm pleased to inform you that your manuscript has been deemed suitable for publication in PLOS ONE. Congratulations! Your manuscript is now with our production department. 

Kind regards, 

on behalf of

Dr. Karel Sedlar 

Academic Editor

PLOS ONE